# Evidence for a finite-momentum Cooper pair in tricolor *d*-wave superconducting superlattices

T. Asaba [1] ✉, M. Naritsuka [1,2], H. Asaeda[1], Y. Kosuge[1], S. Ikemori[1], S. Suetsugu [1], Y. Kasahara [1], Y. Kohsaka [1], T. Terashima[1], A. Daido[1], Y. Yanase[1] & Y. Matsuda [1] ✉

Fermionic superfluidity with a nontrivial Cooper-pairing, beyond the conventional Bardeen-Cooper-Schrieffer state, is a captivating field of study in quantum many-body systems. In particular, the search for superconducting states with finite-momentum pairs has long been a challenge, but establishing its existence has long suffered from the lack of an appropriate probe to reveal its momentum. Recently, it has been proposed that the nonreciprocal electron transport is the most powerful probe for the finite-momentum pairs, because it directly couples to the supercurrents. Here we reveal such a pairing state by the non-reciprocal transport on tricolor superlattices with strong spin-orbit coupling combined with broken inversion-symmetry consisting of atomically thin *d*-wave superconductor $CeCoIn_5$. We find that while the second-harmonic resistance exhibits a distinct dip anomaly at the low-temperature (*T*)/high-magnetic field (*H*) corner in the *HT*-plane for **H** applied to the antinodal direction of the *d*-wave gap, such an anomaly is absent for **H** along the nodal direction. By carefully isolating extrinsic effects due to vortex dynamics, we reveal the presence of a non-reciprocal response originating from intrinsic superconducting properties characterized by finite-momentum pairs. We attribute the high-field state to the helical superconducting state, wherein the phase of the order parameter is spontaneously spatially modulated.

A fundamental assumption of the Bardeen–Cooper–Schrieffer (BCS) theory of superconductivity is that two electrons form a Cooper pair with zero center-of-mass momentum. Realizing exotic superconducting states with finite-momentum pairs that violate this assumption has been a long-sought goal in condensed matter physics. Such a superconducting state is an enticing theoretical possibility but has proven a severe experimental challenge. This is not only because the conditions under which such a superconducting state can be formed are rather stringent but also because smoking-gun experiments to confirm its existence have still been lacking.

Helical superconductivity, in which the amplitude of the superconducting order parameter is constant, but its phase is spontaneously and spatially modulated, has been proposed as a prominent example of such a finite-momentum pair state[1–4]. The realization of helical superconductivity requires a strong Rashba effect that appears as a combined consequence of significant spin-orbit interaction (SOI) and spatial inversion symmetry breaking. When the crystal structure lacks a center of inversion, the SOI may dramatically change the electronic properties, leading to nontrivial quantum states. The key microscopic ingredient in understanding the physics of such non-centrosymmetric materials is the appearance of antisymmetric SOI of

[1]Department of Physics, Kyoto University, Kyoto 606-8502, Japan. [2]Present address: RIKEN Center for Emergent Matter Science, Wako, Saitama 351-0198, Japan. ✉e-mail: asaba.tomoya.4t@kyoto-u.ac.jp; matsuda@scphys.kyoto-u.ac.jp

the single electron states. Asymmetry of the potential in the direction perpendicular to the two-dimensional (2D) plane $\nabla V \| (001)$ induces Rashba type SOI, $\alpha_R \mathbf{g}(\mathbf{k}) \cdot \boldsymbol{\sigma} \propto (\mathbf{k} \times \nabla V) \cdot \boldsymbol{\sigma}$, where $\alpha_R$ is the Rashba coupling, $\mathbf{k}$ is the wave number, $\mathbf{g} = (-k_y, k_x, 0)/k_F$ with $k_F$ the Fermi wave number, and $\boldsymbol{\sigma}$ is the Pauli matrix[5]. Rashba SOI splits the Fermi surface into two sheets with different spin structures. The energy splitting is given by $\alpha_R$, and the spin direction is tilted into the plane, rotating clockwise on one sheet and anticlockwise on the other.

The Rashba SOI has profound consequences on the superconducting states[6,7]. For example, parity is generally no longer a good quantum number, leading to exotic states with a mixture of spin-singlet and spin-triplet components. When the Rashba splitting becomes sufficiently larger than the superconducting gap energy, it has been theoretically proposed that an even more fascinating superconducting state may emerge in 2D superconductors by applying strong parallel magnetic fields; a conventional BCS state with zero-momentum pairs ($\mathbf{k}\uparrow, -\mathbf{k}\downarrow$) formed within spin-textured Fermi surfaces (Fig. 1a) changes into a superconducting state with finite-momentum pairs formed within each spin nondegenerate Fermi surface[1–4] (Fig. 1b). Such a superconducting state appears as a result of the shift of the Rashba-split Fermi surfaces by external parallel fields. When the magnetic field is applied parallel to $\hat{\mathbf{x}}$ axis ($\mathbf{H} = H\hat{\mathbf{x}}$), the center of the two Fermi surfaces with different spin helicity are shifted along $\hat{\mathbf{y}}$ axis in opposite directions. This state, referred to as a helical superconducting state, is characterized by the formation of Cooper pairs ($\mathbf{k} + \mathbf{q_R}\uparrow, -\mathbf{k} + \mathbf{q_R}\downarrow$), where $\mathbf{q}_R = \mu_B H \hat{\mathbf{y}} / \sqrt{\alpha_R^2 + \frac{2E_F}{m}}$ with Bohr magneton $\mu_B$, Fermi energy $E_F$ and quasiparticle mass $m$. This pair formation leads to a state in which the magnitude of the superconducting order parameter is constant, while its phase rotates in space with period $\pi/|\mathbf{q}_R|$ as $\Delta(\mathbf{r}) = \Delta_0 e^{2i\mathbf{q_R} \cdot \mathbf{r}}$.

We note that the helical state is essentially different from the Fulde–Ferrell (FF) and Larkin–Ovchinnikov (LO) states, in which finite-momentum Cooper pairs are formed between sections of the Zeeman-split Fermi surfaces[8,9] (Fig. 1c). A potential FF or LO state has been reported in several candidate materials, by showing a phase transition inside the superconducting state[10,11] through the measurements of magnetization[12], specific heat[13–15], nuclear magnetic resonance[16–18], thermal conductivity[19], ultrasound[20,21], and scanning tunneling microscope[22]. In the FF state, the finite-momentum Cooper pairs lead to the phase modulation of the superconducting order parameter, which is difficult to detect directly. In the LO state, the spatial modulation of the superconducting order parameter due to such pairs gives rise to periodic nodal planes in the crystal. However, it should be emphasized that no direct evidence showing such periodic nodes has been reported so far. This is mainly due to the inherent challenge in directly measuring the momentum of Cooper pairs within a

superconducting state, calling for a novel probe to investigate the Cooper pair momentum.

Very recently, it has been theoretically proposed that superconducting states with finite-momentum Cooper pairs exhibit a current-direction-dependent critical current, namely the superconducting diode effect[23–26]. This diode effect appears due to the nonreciprocal nature of the pair momentum-dependence of the free energy. In the framework of superconductivity, the superconducting current, denoted as $\mathbf{j}$, and the center-of-mass momentum of Cooper pairs, represented by $\mathbf{q}$, are coupled within the free energy expression as $F \sim \mathbf{j} \cdot \mathbf{q}$. This coupling suggests that, analogous to how a magnetic field serves as a probe for investigating ferromagnetic properties, the superconducting current provides a robust means to probe the dynamics of the center-of-mass momentum of Cooper pairs. Notably, the diode effect is significantly enhanced upon entering the helical superconducting state both in $s$-wave[24,26] and $d$-wave superconductors[27]. The enhancement naturally leads to characteristic behaviors of nonreciprocal electron transport (NRET) in general. Therefore, measurements of the NRET provide a powerful tool for revealing the helical superconductivity. The resistance of a 2D film can be described as $R = R_0(1 + \gamma \mu_0 \mathbf{H} \times \hat{\mathbf{z}} \cdot \mathbf{I})$, where $R_0$ and $\mathbf{I}$ are the resistance in the zero-current limit and an electric current, respectively. The coefficient $\gamma$ gives rise to different resistance for rightward and leftward electric currents and can be finite in non-centrosymmetric materials. Unless the resistive transition in magnetic fields is very sharp due to strong pinning, the NRET response can be obtained by measuring the second harmonic resistance $R_{2\omega}$. The comparison between $R_{2\omega}$ at low frequencies in the DC limit and the differential in the critical current has been well-documented across various systems, and the general consensus is that if one is finite, the other will also be finite[28,29].

NRET has been studied in several superconductors[28,30,31]. However, it remains an arduous task to discern whether the observed NRET response stems from intrinsic superconducting phenomena, such as exotic pairing states that contain finite-momentum Cooper pairs. This is because the NRET response can also arise from extrinsic effects such as asymmetric vortex pinning at the edge, surface, and interface, the ratchet effect of the pinning center, and geometry-dependent Meissner shielding effects[32–35].

To address this challenge, we have developed tricolor Kondo superlattices composed of atomically thin $CeCoIn_5$ layers. We have methodically isolated the intrinsic superconducting effects, thereby eliminating extrinsic influences. This approach has unveiled a nonreciprocal response that stems from the intrinsic superconducting properties characterized by the formation of finite-momentum pairs. We associate the observed high-field state with the helical superconducting state.

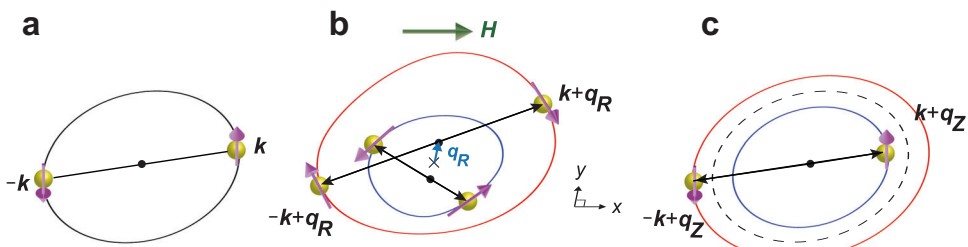

**Fig. 1 | Schematic of the various types of Cooper pairings. a** Conventional BCS pairing state. Zero-momentum pairing with ($\mathbf{k}\uparrow, -\mathbf{k}\downarrow$) occurs between electrons in states with opposite momentum and opposite spins. **b** Helical superconducting state. Arrows on the Rashba-split Fermi surfaces indicate spins. **H** parallel to $\hat{\mathbf{x}}$ axis shifts the center of the small and large Fermi surfaces by $\mathbf{q}_R$ along $+y$ and $-y$ directions, respectively. Pairs are formed within each Rashba-split Fermi surface between the states of ($\mathbf{k} + \mathbf{q_R}\uparrow, -\mathbf{k} + \mathbf{q_R}\downarrow$), leading to a gap function with modulation of phase $\Delta(\mathbf{r}) \propto \exp(2i\mathbf{q_R} \cdot \mathbf{r})$. Cooper-pairs have finite center-of-mass momentum $\mathbf{q}_R$. **c** FF and LO pairing states. Pairing with ($\mathbf{k} + \mathbf{q_Z}\uparrow, -\mathbf{k} + \mathbf{q_Z}\downarrow$) occurs between sections of the Zeeman-split Fermi surfaces, where $\mathbf{q}_Z \approx 2\mu_B H/\hbar v_F$. Cooper-pairs have finite center-of-mass momentum $\mathbf{q}_Z$. In the FF state, the order parameter varies as $\Delta(\mathbf{r}) \propto \exp(2i\mathbf{q_Z} \cdot \mathbf{r})$, while in the LO state, it varies as $\Delta(\mathbf{r}) \propto \cos(2\mathbf{q_Z} \cdot \mathbf{r})$.

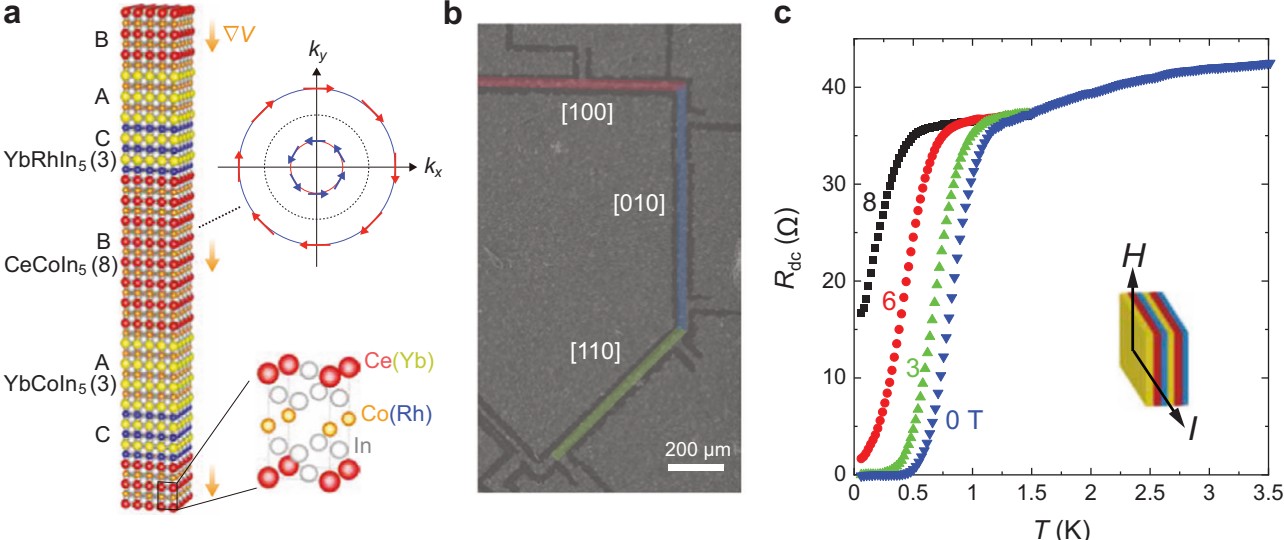

**Fig. 2 | Tricolor *d*-wave superconducting superlattices. a** Schematic representation of noncentrosymmetric tricolor Kondo superlattices with $\cdots A/B/C/A/B/C$ $\cdots$ structure. The sequence of YbCoIn$_5$(3)/CeCoIn$_5$(8)/YbRhIn$_5$(3) is stacked repeatedly 30 times, so that the total thickness is about 300 nm. The orange arrows represent the asymmetric potential gradient $\nabla V$, which gives rise to the Rashba splitting of the Fermi surface with different spin structures. The crystal structure of Ce(Yb)Co(Rh)In$_5$ is also illustrated. **b** The scanning electron microscopy image of a tricolor superlattice patterned by focused ion-beam (FIB). The black line regime corresponds to the area cut by FIB. Red, blue, and green lines indicate the current path along [100], [010], and [110], respectively. The width of the current path is $20 \pm 2\,\mu$m. **c** The temperature dependence of the dc-resistance $R_{dc}$ for **H**||[100] and **I**|| [010].

## Results

### Sample characterization

CeCoIn$_5$ is a well-known heavy-fermion superconductor with the highest bulk $T_c$ of 2.3 K, in which $d_{x^2-y^2}$ superconducting gap symmetry is well established[36,37]. Recently, using state-of-the-art molecular beam epitaxy (MBE) technology[38], we have successfully fabricated epitaxial thin films of CeCoIn$_5$ and atomic-thickness superlattices of CeCoIn$_5$, alternating with layers of different materials[39,40] such as CeIn$_3$[41], CeRhIn$_5$[42], YbRhIn$_5$[43], and YbCoIn$_5$[44,45]. The epitaxial growth of these superlattices was confirmed by reflection high energy electron diffraction (RHEED). The interfaces between CeCoIn$_5$ and adjacent layers are characterized by their distinct atomic sharpness, a feature confirmed through multiple methods. Specifically, both transmission electron microscopy (TEM) and electron energy loss spectroscopy (EELS) have substantiated this attribute[42]. In our TEM analyses, we verified the absence of texturing over micrometer-scale areas along the layer direction. Furthermore, scanning tunneling microscopy (STM) topographic images corroborate this finding; they reveal an atomically flat surface of the CeCoIn$_5$ layers[46,47], with no indications of texturing. This collective evidence strongly supports that the samples are indeed single-crystalline and devoid of any textural anomalies.

The bulk CeCoIn$_5$ possesses the inversion center. Then, fabricating tricolor superlattices with an asymmetric $\cdots A/B/C/A/B/C\cdots$ arrangement, in which non-superconducting metals sandwich CeCoIn$_5$ with atomic layer thickness, we can introduce the global inversion symmetry breaking (Fig. 2a)[39,40,43,48]. Given that this superlattice comprises three distinct materials, it will be designated as tricolor henceforth. This tricolor system provides an ideal platform for revealing the helical superconducting state for the following reasons. First, Ce atoms have a large SOI, and the condition that the Rashba-SOI well exceeds the superconducting gap has been confirmed in various superlattices of CeCoIn$_5$, including the present tricolor superlattice, by the highly enhanced upper critical field from Pauli limited critical fields in bulk (see SI and ref. 40). Second, Cooper pairs can be confined in atomic CeCoIn$_5$ layers, forming 2D superconductivity[48]. Third is the strong electron correlation effect in CeCoIn$_5$. It has been theoretically pointed out that the correlation further strengthens the effect of Rashba SOI[49].

Furthermore, the suppression of the orbital pair-breaking effects promotes the appearance of helical superconducting phases. These features make the CeCoIn$_5$ superlattice system unique and suitable for realizing helical superconductivity compared to weakly correlated systems. Finally, *d*-wave superconductors are expected to respond differently to in-plane magnetic fields directed for the nodal and antinodal directions, possibly allowing the intrinsic NRET to be extracted by changing the field direction.

The tricolor superlattices with a *c*-axis-oriented structure have been epitaxially grown on an MgF$_2$ substrate using the MBE technique. The structure consists of alternating layers: 3-unit-cell-thick (3-UCT) YbCoIn$_5$, 8-UCT CeCoIn$_5$, and 3-UCT YbRhIn$_5$. In this configuration, YbCoIn$_5$ and YbRhIn$_5$ act as conventional non-superconducting metals (Fig. 2a). The deposition sequence of YbCoIn$_5$/CeCoIn$_5$/YbRhIn$_5$ was repeated 30 times. Within these tricolor superlattices, none of the layers act as mirror planes, thereby introducing broken inversion symmetry along the stacking direction. It is noteworthy that the present superlattice structure, comprising 30 layers, significantly reduces the influence of the substrate, rendering it negligible for our experimental purposes. In addition, for comparison, we fabricated bicolor superlattices with $\cdots A/B/A/B\cdots$ stacking structure, in which spatial inversion symmetry is preserved. In this superlattice, the deposition sequence of YbCoIn$_5$ (5-UCT)/CeCoIn$_5$ (8-UCT) was repeated 30 times.

Given the necessity for a precise in-plane application of the magnetic field in this study, we employed the 8-UCT CeCoIn$_5$ tricolor superlattice sample previously characterized in ref. 48. This referenced work extensively investigated the temperature and angular dependence of the upper critical field for the sample. Since CeCoIn5 is a quasi-2D superconductor with a larger coherence length in the *ab*-plane ($\xi_{ab}$) compared to along the *c*-axis ($\xi_c$), and the thickness of 8-UCT CeCoIn$_5$ layers in the present superlattices is comparable to $\xi_c$, elliptical vortices are likely to form within these layers when exposed to a parallel field. Moreover, the presence of the strong Rashba SOI is confirmed by the suppressed Pauli limit[48]. For the sake of achieving a high current density and ensuring meticulous control over the current orientation, the sample underwent patterning utilizing a focused ion beam (FIB) as depicted in Fig. 2b. We note that both the $T_c$ and upper

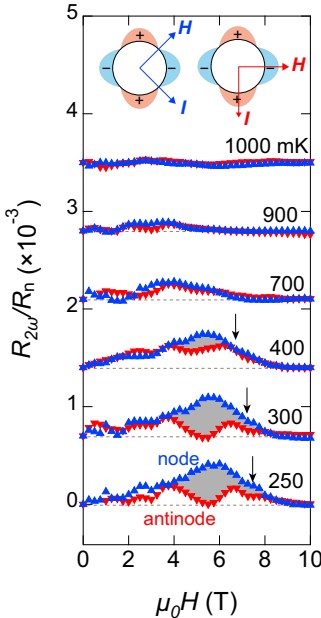

**Fig. 3 | Nonreciprocal electronic transport in the superconducting state of tricolor superlattice.** The field dependence of second harmonic resistance $R_{2\omega}$ normalized by $R_n$ in tricolor superlattice for two configurations. Blue upper triangles show $R_{2\omega}/R_n$ for both **H** and **I** applied parallel to the $d$-wave nodal direction (**H**∥[110], **I**∥[1$\bar{1}$0]), as illustrated by the left panel in the inset. Red lower triangles show $R_{2\omega}/R_n$ for antinodal configuration (**H**∥[100], **I**∥[010], right panel in the inset). The curves are vertically shifted for clarity. The dashed lines indicate the baselines. The gray area represents the difference between the two configurations $\Delta R_{2\omega}/R_n$. Arrows indicate $H_{c2\parallel}$ determined by $R_{dc}$.

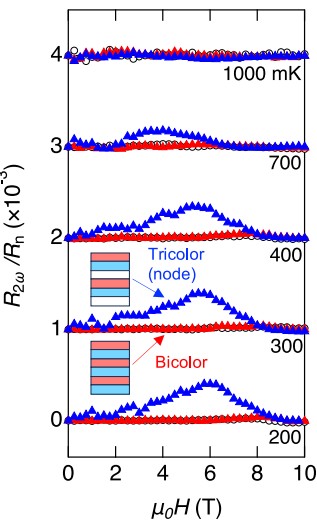

**Fig. 4 | Nonreciprocal electronic transport in the superconducting state of bicolor superlattice.** For the bicolor superlattice, the sequence of YbCoIn$_5$(5)/CeCoIn$_5$(8) is stacked repeatedly for 30 times. The field dependence of second harmonic resistance $R_{2\omega}$ normalized by $R_n$ in bicolor superlattice for two configurations is shown. Red upper triangles show $R_{2\omega}/R_n$ for both **H** and **I** applied parallel to the $d$-wave nodal direction (**H**∥[110], **I**∥[1$\bar{1}$0]). Black open circles show $R_{2\omega}/R_n$ for antinodal configuration (**H**∥[100], **I**∥[010]). For comparison, $R_{2\omega}/R_n$ for nodal configuration is also potted by Blue upper triangles. The curves are vertically shifted for clarity.

critical field of this sample changed only slightly before and after FIB patterning (See Fig. S1). The sample becomes superconducting at $T_c = 0.83$ K defined as the temperature at which the dc-resistance $R_{dc}$ drops to 50% of its normal state value at the onset (Fig. 2c). The observed decrease in $T_c$ of the superlattice, compared to the bulk sample, can plausibly be attributed to the modification of the electronic band structure from 3D to 2D. This alteration leads to changes in the antiferromagnetic fluctuations, which play an essential role in the unconventional superconductivity of CeCoIn$_5$[42]. Nonreciprocal transport measurements are carried out by the standard lock-in technique (see Materials and Methods in SI). The $R_{2\omega}$ curves are antisymmetrized with respect to a magnetic field. The misalignment of **H** from the ab plane is less than 0.05°.

**Nonreciprocal transport**

Figure 3 depicts $R_{2\omega}$ normalized by normal state resistance $R_n$ when both in-plane **H** and **I** are applied to nodal (**H**∥[110], **I** ∥ [1$\bar{1}$0]) and antinodal (**H**∥[100], **I**∥[010]) directions (see the inset). NRET can manifest even in the normal state in the presence of inversion symmetry breaking. However, for both configurations, no discernible NRET is observed in high temperatures or high magnetic fields, indicating that the observed response purely arises from superconducting properties. Therefore, despite the broadening of resistive transition that the inhomogeneity may cause, only the superconducting response of $R_{2\omega}$ can be extracted. For the nodal configuration, $R_{2\omega}$ increases with $H$ at low temperatures, peaking at $\mu_0 H \sim 6$ T and disappearing at high fields. It should be noted that such a single-peak structure as a function of $H$ in the superconducting state has also been observed in NbSe$_2$[31] and ion-gated SrTiO$_3$[30], in which peaks gradually shift and decrease as the temperature elevates. It was found that such a structure can be explained by the vortex motion. On the other hand, for the antinodal configuration, while a similar peak is observed at high

temperatures, the peak is suppressed around $\mu_0 H \sim 5$ T at low temperatures, exhibiting a distinct dip anomaly. While there may be an underlying structure at low magnetic fields, the signal is vanishingly small. Therefore, in this study, our focus is on the signals at high fields.

As noted above, finite $R_{2\omega}$ originates from the Cooper pairs. There are two possible sources for the NRET response in the superconducting state. One is extrinsic origin such as Meissner screening current and vortex motion, and the other is intrinsic origin due to the exotic superconducting state with finite-momentum pairing. We here discuss extrinsic origins. The direction-dependent critical current can be induced by the combination of the Meissner screening effect and the asymmetric vortex surface barrier arising from the sample edges[50]. However, since such an effect is important only at a very low field around the lower critical field, it is negligibly small in the present field range, which significantly exceeds the Pauli limit.

Another extrinsic origin is the asymmetric vortex motion. When both **H** and **I**(⊥**H**) are applied in-plane, the vortices move in and out across the interfaces. If there is an asymmetric vortex pinning potential at the interface of the different materials, NRET may occur. In the tricolor superlattices, different vortex thread pinning potentials on either side of the superconductor interface may induce NRETs of different amplitudes, leading to an asymmetric motion that can generate a net NRET signal. To confirm this, we measured the NRET response in bicolor ⋯$A/B/A/B$⋯ stacking superlattices with canceling contributions on both sides of the interface. Figure 4 depicts the NRET of the bicolor superlattice for the nodal (**H**∥[110], **I** ∥ [1$\bar{1}$0]) and antinodal (**H**∥[100], **I**∥[010]) configurations. The figure includes the NRET of the tricolor superlattice in the nodal configuration for comparison. In the bicolor superlattice, the NRET is found to be absent or negligibly small, especially when compared to that of the tricolor superlattice. This supports the presence of the NRET response arising from vortex motion perpendicular to the layers. To rule out the possibility of pancake vortices perpendicular to the layer created by small but finite misalignment of **H** out of the 2D plane inducing the NRET effect, we measured the NRET by applying **H** tilted about 4° from the $ab$ plane and found no such effect in the tricolor superlattices (See Fig. S4).

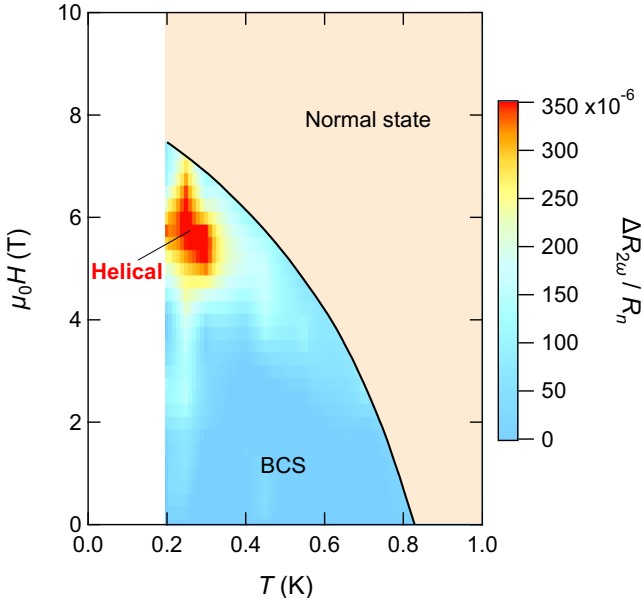

**Fig. 5 | Superconducting phase diagram of the tricolor superlattice determined by nonreciprocal electron transport properties.** The solid line is $H_{c2\parallel}$ determined by $R_{dc}$. $H_{c2\parallel}$ line for nodal and antinodal directions well coincides each other. The light brown area represents the normal regime. In the superconducting state, the difference of $R_{2\omega}/R_n$ between two configurations, $\Delta R_{2\omega}/R_n$, is plotted in color. The blue area at low fields represents the BCS regime, where $\Delta R_{2\omega}/R_n$ is negligibly small while finite $R_{2\omega}$ is observed for both configurations due to extrinsic contributions from vortex motion. The red area at high fields corresponds to the helical superconducting state, where the finite $\Delta R_{2\omega}/R_n$ appears due to the intrinsic contribution originating from Cooper pairs.

Additionally, no NRET effect was observed in the Lorentz force-free geometry, $\mathbf{H}\parallel\mathbf{I}$. These results indicate that the extrinsic NRET response, if present, arises from asymmetric vortex motion perpendicular to the layers.

We note that in the present tricolor superlattices, the vortex motion perpendicular to the 2D plane is almost independent of the in-plane directions of $\mathbf{H}$ and $\mathbf{I}(\perp\mathbf{H})$. This can be attributed to the following reasons: First, the superlattices with tetragonal crystal symmetry have no twin boundaries. Additionally, as illustrated in Fig. S1, the in-plane critical field $H_{c2}$ is almost isotropic. The electronic structure of a vortex is determined by its coherence length, which, in turn, is determined by the average of the Fermi velocity. Given that the coherence length is directly related to $H_{c2}$, an isotropic in-plane $H_{c2}$ suggests that the electronic structure of vortices for $\mathbf{H}\parallel[100]$ is very similar to that for $\mathbf{H}\parallel[110]$. This isotropy in $H_{c2}$ is also reflected in the behavior of the electrical current, which is dependent on the Fermi velocity. Consequently, the uniformity of the average Fermi velocity within the plane implies an isotropic current contribution as well.

To separate the intrinsic contribution from the extrinsic one, we take the difference between two configurations, as represented by the gray area in Fig. 3. Notably, except for the gray area at low temperatures, $R_{2\omega}$ from the two configurations nearly overlaps. This indicates that $R_{2\omega}$ for both configurations is dominated by extrinsic vortex motion except for the regime where $R_{2\omega}$ exhibits a dip anomaly at low temperatures around $\mu_0 H \sim 5\,\text{T}$ for the antinodal configuration. Therefore, the dip anomaly when both $\mathbf{H}$ and $\mathbf{I}$ are applied to antinodal directions is attributed to an intrinsic origin arising from the Cooper pairs superimposed on the extrinsic vortex contribution. To obtain further information on the origin of the dip, $R_{2\omega}$ was measured at different relative angles of $\mathbf{H}$ and $\mathbf{I}$; $\mathbf{H}\parallel[100]$ and $I\parallel[1\bar{1}0]$, and $\mathbf{H}\parallel[110]$ and $\mathbf{I}\parallel[010]$ (see SI for details). The results show that the appearance of the dip anomaly is determined by field direction, not by the current

direction, implying that the dip anomaly is related to the superconducting gap structure.

## Phase diagram

The NRET response provides pivotal information on the superconducting phase diagram of the tricolor superlattice displayed in Fig. 5. The solid line in Fig. 5 represents the upper critical field for $\mathbf{H}\parallel[100]$, $H_{c2\parallel}$, determined by $H$ where $R_{dc}$ reaches 50% of $R_n$. We find that $H_{c2\parallel}$ line for $\mathbf{H}$ applied nodal direction well coincides with that for antinodal direction, indicating a similar $HT$-phase diagram (Fig. S1). The upper right (colored in light brown) area in Fig. 5 represents the normal state. In the superconducting state below $H_{c2\parallel}$, the difference of $R_{2\omega}$ between two configurations, $\Delta R_{2\omega} \equiv R_{2\omega}(\mathbf{H}\parallel[110], \mathbf{I}\parallel[1\bar{1}0]) - R_{2\omega}(\mathbf{H}\parallel[100], \mathbf{I}\parallel[010])$, normalized by $R_n$, is plotted in color; the gray area displayed in Fig. 3 corresponds to $\Delta R_{2\omega}/R_n$. In the light blue area at low fields in Fig. 5, while finite $R_{2\omega}$ is observed for both configurations due to extrinsic contributions from vortex motion, $\Delta R_{2\omega}$ is negligibly small. In the red area at high fields, the finite $\Delta R_{2\omega}$ appears due to the intrinsic contribution originating from Cooper pairs. Note that as shown by arrows in Fig. 3 indicating $H_{c2\parallel}$ determined by $R_{dc}$, $\Delta R_{2\omega}$ vanishes at around $H_{c2\parallel}$, while small but finite $R_{2\omega}$ remains at $H_{c2\parallel}$, likely due to the superconducting fluctuation effect and inhomogeneity.

In the bicolor system where the NRET is not observed, no discernible difference is observed between the antinodal and nodal configurations (Fig. 4). The fact that the bicolor superlattice preserves global inversion symmetry leads us to conclude that the emergence of NRET difference in the tricolor lattice, $\Delta R_{2\omega}$, is an intrinsic phenomena arising from the Rashba SOI. The intrinsic NRET emerges as a direct consequence of the state with finite-momentum pairs, and such an effect is negligibly small in the BCS state. Therefore, the results of Fig. 5 provide evidence for the appearance of a high-field superconducting state at the low-$T$/high-$H$ corner, distinct from the low-field BCS state. Although the anomalous upturn behavior of $H_{c2\parallel}$ at low temperatures has been suggested in the previous study[48], the superconducting state at high fields had remained an unresolved issue, including the possible existence of a new phase. We note that we can rule out the possibility that the observed nonreciprocal phenomena are tied to the so-called $Q$-phase[51], in which the superconductivity may be intertwined with magnetic order, in bulk $CeCoIn_5$ for the following reasons. Firstly, in the basic Drude model, nonreciprocal transport is independent of spin. Then, the primary effect of the $Q$-phase on nonreciprocal transport is the Brillouin zone folding, but this has a negligible effect. In addition, in the $Q$-phase, where spatial modulations of order parameters appear, electron scattering should increase, resulting in the suppression of the nonreciprocal response. However, our observations indicate the opposite in the current case. Furthermore, Cooper pairs in the $Q$-phase do not carry a finite momentum. Hence, even when the inversion symmetry is broken, the momentum of the Cooper pair remains unchanged.

As discussed above, the NRET is caused purely by the supercurrents, which directly couple to the finite momentum Cooper pairs. It is crucial to clarify why the nonreciprocal signal is strongly suppressed in the normal state, despite the presence of broken inversion symmetry. This suppression can be attributed to the fact that the signal is proportional to $1/E_F$ in the normal state and to $1/\Delta$ in the superconducting state, where $E_F$ and $\Delta$ represent the Fermi energy and the superconducting gap, respectively. Consequently, the signal is significantly smaller in the normal state[52,53].

In the context of magnetic field-induced resistive transitions in superconductors, particularly when the transition is broadened due to superconducting fluctuations as observed in this study, there is a degree of uncertainty in selecting the appropriate resistance threshold (0.9 $R_n$, 0.5 $R_n$, or 0.1 $R_n$) to define the mean-field upper critical field. However, given that NRET is vanishingly small in the normal state, it is

natural to associate the mean-field upper critical field with the emergence point of observable NRET. In the present case, the magnetic field corresponding to $0.5\,R_n$ closely aligns with this emergence point. Consequently, this value has been adopted to define the mean-field upper critical field. It is worth noting that the intrinsic contribution of NRET predominantly becomes discernible in the resistance range between $R = 0.1R_n$ and $R = 0.5\,R_n$ as shown in Fig. S5.

## Discussion

It should be emphasized that we can discard the possibility of both FF and LO states[10] (Fig. 1c), where pairing between sections of the Zeeman-split Fermi surfaces results in Cooper-pairs ($\mathbf{k} + \mathbf{q}_Z\uparrow$, $-\mathbf{k} + \mathbf{q}_Z\downarrow$) with momentum $\mathbf{q}_Z \approx \mu_B \mathbf{H}/\hbar v_F$ ($v_F$ is the Fermi velocity), as the origin of the intrinsic NRET phenomena. This is because the energy of the Rashba spin splitting is overwhelmingly larger than the superconducting gap energy in the present tricolor superlattices, as demonstrated in ref. 48 (see Fig. S2). In this situation, FF- and LO-type pair formation cannot occur. In the absence of inversion symmetry, the LO phase can be characterized by a general order parameter, formulated $ae^{2iq_{Zr}} + be^{-2iq_{Zr}}$, $a \neq b$. This phase, as defined by the order parameter, is commonly referred to as the "stripe phase". However, theoretical predictions suggest that the stripe phase only emerges within a narrow region at low temperatures in the $HT$-phase diagram[2], whereas helical superconductivity manifests within a more expansive phase region surrounding it. Considering this, it seems plausible that our experimental observations represent helical superconductivity. It may be possible, however, to observe a stripe phase at even lower temperatures. Based on these results, we conclude that the high field regime indicated by the red color in Fig. 5 represents the helical superconducting state, and the low field regime by light blue corresponds to the BCS state. Although it is theoretically challenging to estimate quantitatively the momentum of Cooper pairs from experimental results, we can estimate the NRET values in the fluctuation regime near the upper critical field. The theoretical estimate[54] of $10^{-17}\,\Omega\mathrm{m}^3/\mathrm{A}$ shows in good agreement with the experimentally observed $5 \times 10^{-18}\,\Omega\mathrm{m}^3/\mathrm{A}$. This strengthens the conclusion of our study.

The strong field-orientation dependence of intrinsic NRET likely appears as a result of the direction-dependent Doppler shift of the quasiparticles in $d$-wave superconductors. When $\mathbf{H}$ is applied parallel to the nodal direction, quasiparticles around the nodes perpendicular to the magnetic field are excited. When the current is applied parallel to these nodes, the system exhibits more metallic behavior compared with that for the antinodal direction. Although such a simple interpretation should be scrutinized, the present results also point toward the importance of nodal structure for the direction-dependent NRET. We note that the finite-momentum pairing state has been suggested in the pair-density-wave (PDW) state in the pseudogap phase of cuprates by scanning tunneling microscope measurements[55]. Therefore, it is highly intriguing to apply the present direction-dependent NRET to the putative PDW state.

The NRET effect arising from the intrinsic superconducting response observed in the tricolor $d$-wave superconducting superlattice with strong Rashba interaction provides evidence for the emergence of a superconducting state with finite-momentum Cooper pairs at high fields, most likely a helical superconducting state. Such a unique state provides a platform to investigate the novel fermionic superfluid systems beyond the BCS pairing states.

## Methods
### Device fabrication
The tricolor Kondo superlattices of $YbCoIn_5/CeCoIn_5/YbRhIn_5$ were epitaxially grown on $MgF_2$ substrates by the molecular beam epitaxy (MBE) technique. Firstly, a $CeIn_3$ layer was grown as a buffer layer on

the $MgF_2$ substrate, then the 3-unit-cell-thick (3-UCT) $YbCoIn_5$, 8-UCT $CeCoIn_5$, and 3-UCT $YbRhIn_5$ layers are alternatively stacked for 30 times. The total thickness of the superlattice is ~ 300 nm. The quality of the superlattices was checked by multiple techniques, including X-ray diffraction measurements. For detailed information on the sample growth and characterization, see Ref. (28). The superlattices were then patterned using the focused ion beam (FIB) technique (JEOL, JIB-4501). The current path was carefully aligned along [100], [110], [010], and [1$\bar{1}$0] directions. A Ga source was used to mill the sample. The width of the FIB-cut sample is ~ 20 $\mu$m. The standard silver paste was used to make electronic contacts.

### Nonreciprocal transport
We measured the first and second harmonic resistance of superlattices by using an AC current source (Keithley, 6221) and AC lock-in amplifiers (Stanford Research Systems, SR830) in a dilution refrigerator. The current of 20 $\mu$A was applied to the sample. A frequency of 13.7 Hz was used. The dc-resistance was measured by the first harmonic resistance, $R_{dc} = R_\omega$. The Cernox thermometers with magnetic-field calibration were used to control the sample temperature. The in-plane magnetic field was precisely applied by using a built-in rotator. The sample temperature was carefully monitored so that the current did not heat the sample. The second harmonic resistance was antisymmetrized as a function of a magnetic field. To obtain the antisymmetrized component of $R_{2\omega}$ from the raw data, we calculated $R_{2\omega}(H) = \frac{R_{2\omega}^{\mathrm{raw}}(H) - R_{2\omega}^{\mathrm{raw}}(-H)}{2}$.

## Data availability
All data were presented in the Article and the Supplementary Information. Source data are provided with this paper.

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

## Acknowledgements

A part of this work was supported by PRESTO (No. JPMJPR2252; T.A.) and CREST (No. JPMJCR19T5; Y.M.) from Japan Science and Technology (JST), Grants-in-Aid for Scientific Research (KAKENHI) (Nos. 18H05227, 18H03680, 18H01180, and 21K13881) and Grant-in-Aid for Scientific Research on innovative areas "Quantum Liquid Crystals" (No. JP19H05824) from Japan Society for the Promotion of Science (JSPS).

## Author contributions

T.A. and Y.M. conceived and supervised the study. T.A., H.A., Y.Kos., S.I., S.S., and Y. Koh performed transport measurements. M.N., Y.Ka., and T.T. synthesized samples. Theoretical discussions were provided by A.D. and Y.Y. All authors discussed the results and contributed to writing the manuscript.

## Competing interests

The authors declare no competing interests.

## Additional information

**Supplementary information** The online version contains
supplementary material available at

T. Asaba or Y. Matsuda.

**Peer review information** *Nature Communications* thanks Noah Yuan and
the other, anonymous, reviewer(s) for their contribution to the peer
review of this work. A peer review file is available.

