## [Peer Review File · Nature Communications]

Evidence for a finite-momentum Cooper pair in tricolor d-wave superconducting superlatticesReviewer #1 (Remarks to the Author):

In their manuscript entitled "Evidence for a finite-momentum Cooper pair in tricolor d-wave superconducting superlattices" T. Asaba et al. have investigated YbCoIn₅/CeCoIn₅/YbRhIn₅ tri layer superstructure to elucidate the existence of the finite momentum cooper pairing. The study utilized second harmonic measurements while applying current and in-plane magnetic fields in different directions. While the paper is well-written and based on sound experimental reports, the significance and novelty of the work warrant careful consideration for publication in a high-impact journal. Nevertheless, there are some issues I would appreciate the authors considering as below.

1. The evidence of finite momentum pairing is not unprecedented, with numerous reports affirming its existence. Therefore, the utility of this new technique remains unclear. It would be beneficial to ascertain if it can quantitatively calculate the magnitude of the finite momentum, adding an intriguing dimension to the study.
2. The role of the tri-color structure requires a more comprehensive discussion. (a) Investigating whether a single-layer sample exhibits a similar effect is essential, and measurements on a single-layer sample should be explored. (b) The impact of thickness variation in the tri-layer structure, especially regarding the formation of elliptical vortices, warrants elucidation. Why is the tri-layer structure necessary, and how does the presence of the substrate itself not break inversion symmetry?
3. A more detailed crystallographic analysis of the tri-layer heterostructure is needed. Previous studies by the authors (ref 40) showcased good TEM data, but the RHEED data was not convincing. Clarification on how the authors determined the single crystalline nature of the tri-layer structure and its lack of texturing is crucial, given the substantial device size of the FIB-fabricated sample.
4. Inclusion of RBS data on stoichiometry would be insightful, particularly considering the lower T_c of the Tri-color film compared to stoichiometric CeCoIn₅. Exploring why the T_c is reduced in tri-layer films could provide valuable insights.
5. The conclusion that the asymmetric vortex motion perpendicular to the 2D plane is independent of in-plane current and magnetic fields needs further explanation. Why does vortex motion not vary in different crystallographic directions?
6. In Fig. 3, the reason behind the changes in peak position (R_{2w}/R_n) with different temperatures should be discussed. The reasons behind multiple oscillations in the antinode and less prominent oscillation in the node, especially at lower temperatures, should be addressed.
7. Figure S5 in the supplementary information compares the non-reciprocal effect between tricolor and bicolor samples for specific configurations (when the magnetic field and the current are applied parallel to the nodal direction ($H \parallel [110]$, $I \parallel [1-10]$)). It would be beneficial to extend this comparison to the antinodal configuration ($H \parallel [100]$, $I \parallel [010]$) for a more comprehensive understanding.

Reviewer #2 (Remarks to the Author):

This manuscript describes the measurements on the second harmonic generation of the superconducting lattices, with a nice theory for the observed results. However, one puzzle prevents me from recommending the publication of the current form of this manuscript. According to my understanding, measurements on the second harmonic generation should be done when the resistance is finite, namely in the fluctuating regime of superconductivity. It occurs to me that the upper critical field here is determined by $0.5R_n$ criterion where normal state does not completely vanish. I am wondering, what would be H_{c2} when the criterion is 0 or $0.1R_n$? I would like to know how much the contribution of normal state is, since the normal state already breaks inversion symmetry and is sufficiently able to generate second harmonics.

We would like to thank all the reviewers for their meticulous and thoughtful review of our manuscript. Their constructive feedback has been instrumental in enhancing the clarity and strength of our work. In the following sections, we address each of the reviewers' comments, which are presented in purple for ease of reference. Additionally, we have made revisions to the manuscript based on these comments, with all changes clearly highlighted in red.

Reviewer #1 (Remarks to the Author):

In their manuscript entitled “Evidence for a finite-momentum Cooper pair in tricolor d-wave superconducting superlattices” T. Asaba et al. have investigated YbCoIn₅/CeCoIn₅/YbRhIn₅ tri layer superstructure to elucidate the existence of the finite momentum cooper pairing. The study utilized second harmonic measurements while applying current and in-plane magnetic fields in different directions. While the paper is well-written and based on sound experimental reports, the significance and novelty of the work warrant careful consideration for publication in a high-impact journal. Nevertheless, there are some issues I would appreciate the authors considering as below.

First, we thank the referee for spending valuable time in reviewing our manuscript and providing detailed comments.

1. The evidence of finite momentum pairing is not unprecedented, with numerous reports affirming its existence. Therefore, the utility of this new technique remains unclear. It would be beneficial to ascertain if it can quantitatively calculate the magnitude of the finite momentum, adding an intriguing dimension to the study.

(1) On previous finite-momentum-pairing experiments:

As mentioned by the reviewer, finite momentum pairing has been discussed in so-called Fulde-Ferrell-Larkin-Ovchinnikov (FFLO) state in which Cooper pairs are formed between Zeeman-split Fermi surfaces. In this context, many studies have reported the data showing the emergence of a field-induced phase at low temperatures. (We note that some of the present authors have extensively studied the FFLO state in various materials, using methods such as thermodynamics, thermal transport, ultrasound, NMR, and STM, and published many papers on experimental and theoretical results, including review papers.) Indeed, some of the reported field-induced superconducting state may be the FFLO state. However, in our stance, it is necessary to provide evidence for finite-momentum pairing to establish the FFLO pairing state ($\mathbf{k}, -\mathbf{k}+\mathbf{q}$) rather than the BCS pairing state ($\mathbf{k}, -\mathbf{k}$). We stress that despite tremendous studies over the decades, conclusive experimental evidence for the formation of the finite-momentum Cooper pairs has not been established in the putative FFLO state. In fact, in the LO state, which is more energetically stable than the FF state, finite momentum Cooper pairs give rise to the appearance of the periodic nodal planes in the sample, but such nodal planes have not been experimentally confirmed yet. This makes it a long-standing challenge to ascertain if the Cooper pairs truly possess a finite center-of-mass momentum.

(2) Observing finite-momentum Cooper pairs through nonreciprocal transport:

We stress that it is the supercurrent that most directly couples to the momentum of Cooper pairs. In the framework of superconductivity, the superconducting current, denoted as \mathbf{j} , and the center-of-mass momentum of Cooper pairs, represented by \mathbf{q} , are coupled within the free energy expression as $F \sim \mathbf{j} \cdot \mathbf{q}$. This coupling suggests that, analogous to how a magnetic field serves as a probe for investigating ferromagnetic properties, the superconducting current provides a robust means to probe the dynamics of the center-of-mass momentum of Cooper pairs. Very recently, three theoretical groups (Kyoto, RIKEN, and MIT) have independently drawn attention to nonreciprocal transport in the superconducting state. All three groups have proposed that “intrinsic” nonreciprocal transport in the superconducting state acts as the most direct method to probe the momentum of Cooper pairs. Therefore, if we can differentiate the “extrinsic” contributions of nonreciprocal transport that arise from the vortex motion and Meissner current, we can identify the presence of a finite-momentum pairing state. This is a major advance over the previous experimental results, paving the way for novel investigations into the finite momentum pairing state. It should be emphasized that the FF and LO states can be excluded owing to the large Rashba splitting, and hence helical superconductivity is the only candidate to detect the finite-momentum Cooper pairing through nonreciprocal transport. We emphasized this point in the revised manuscript.

Responding to the referee's comment about the alignment between theoretical predictions and our experimental findings, it is challenging to estimate quantitatively the momentum of Cooper pairs from experimental results at the present stage of theoretical study. Instead, some of our co-authors have recently theorized NRET values in the fluctuation regime near T_c and H_{c2} in arXiv:2302.10677. The theoretical estimate of 10^{-17} Ohm m^3/A shows remarkable alignment with our experimental observation of 5×10^{-18} Ohm m^3/A . This correlation strengthens the credibility of our experimental data, thereby substantiating the conclusions of our study. We explicitly stated this estimation in the revised manuscript. We appreciate this comment by the reviewer.

2. The role of the tri-color structure requires a more comprehensive discussion. (a) Investigating whether a single-layer sample exhibits a similar effect is essential, and measurements on a single-layer sample should be explored. (b) The impact of thickness variation in the tri-layer structure, especially regarding the formation of elliptical vortices, warrants elucidation. Why is the tri-layer structure necessary, and how does the presence of the substrate itself not break inversion symmetry?

(a) We suppose that a 'single layer' refers to a mono-color structure comprising solely CeCoIn₅. In our study, employing a thick single layer of CeCoIn₅ (e.g., 300 nm) is not appropriate for the comparison with the tricolor case. This is because the superconductivity in such thick films is strongly suppressed by the Pauli paramagnetic effect, resulting in much lower H_{c2} compared to the tricolor case. The experiments on thinner single layers of CeCoIn₅, specifically around 8 unit-cell-thick (UCT), would be more direct. In pursuit of this, we attempted to fabricate CeCoIn₅ thin

films with 8 UCT. Unfortunately, these atomically thin films show significant sensitivity to air, making it virtually impossible to perform transport measurements. Our efforts also extended to protecting CeCoIn₅ with a YbCoIn₅ film layer. Unfortunately, YbCoIn₅ also exhibited oxidation, which hindered our ability to precisely control Rashba Spin-Orbit Interaction (SOI). Increasing the thickness of the YbCoIn₅ layer to mitigate this issue resulted in the dominance of transport within this layer, thereby obscuring the signals from the underlying CeCoIn₅.

Given these challenges, we determined that constructing multilayer superlattices was the most effective approach for this study. This method allows for the introduction of Rashba SOI with high controllability while minimizing surface oxidation effects. Consequently, we do believe that a bicolor system with preserved spatial inversion symmetry represents the most direct and suitable control experiment for our research objectives.

(b) We realize that our explanation for the vortex-induced NRET might be confusing. Because CeCoIn₅ is quasi-2D superconductor with coherence length in the ab-plane ξ_{ab} is larger than the c-axis coherence length ξ_c , the vortices are elliptical. This is also the case for our superlattices, where the thickness of CeCoIn₅ layers is comparable to the c-axis coherence length. However, the shape of the vortex does not affect the presence of the vortex contribution in tri-color systems as follows. In tricolor structures, or systems where spatial inversion symmetry is broken, the potential experienced by a vortex when it penetrates into another layer differs at the top and bottom interfaces. This asymmetry leads to a variation in vortex resistance depending on the current direction, thereby inducing NRET. In bicolor structures, however, the potentials at the CeCoIn₅ interfaces are symmetrical, resulting in the cancellation of such effects and the absence of NRET. In the revised manuscript, we have moved the discussion on the shape of the vortex in order to prevent confusion.

Additionally, while the referee correctly points out that similar effects could occur at the interface between the substrate and CeCoIn₅, the design of our study minimizes this influence. Our superlattice structure, comprising 30 layers, significantly reduces the substrate's effect, rendering it negligible for our experimental purposes. This consideration further underscores the rationale behind employing a superlattice structure in our study. We explicitly stated thus in the revised manuscript.

3. A more detailed crystallographic analysis of the tri-layer heterostructure is needed. Previous studies by the authors (ref 40) showcased good TEM data, but the RHEED data was not convincing. Clarification on how the authors determined the single crystalline nature of the tri-layer structure and its lack of texturing is crucial, given the substantial device size of the FIB-fabricated sample.

We wish to emphasize the distinct atomic sharpness of the interface between CeCoIn₅ and adjacent layers, a characteristic confirmed through multiple methods. Specifically, not only TEM but also Electron Energy Loss Spectroscopy (EELS) have substantiated this feature, as illustrated

in Fig. R1. In our TEM analyses, we verified the absence of texturing over micrometer-scale areas along the layer direction. Complementarily, Scanning Tunneling Microscopy (STM) topographic images further corroborate this finding; they reveal an atomically flat surface of CeCoIn₅ layers, with no indications of texturing (Fig. R2). This collective evidence strongly supports our assertion that the sample is indeed single-crystalline and devoid of any textural anomalies. Also, it is important to note RHEED was employed primarily to confirm the epitaxial growth. The single-crystalline nature of the samples, however, was more conclusively established through the integration of the aforementioned experiments. We have described the sample characterization in the revised manuscript.

We also wish to highlight a significant accomplishment of our research group: we are the pioneers in successfully growing epitaxial thin films of the heavy fermion superconductor CeCoIn₅, a feat achieved through MBE techniques. This process is particularly challenging due to the intricate nature of working with rare-earth elements. Our successful synthesis of tricolor superlattices, maintaining atomic layer precision, is an achievement we believe warrants recognition, considering the complexity and challenges inherent in this endeavor.

We sincerely hope the reviewer acknowledges these unique challenges and the innovative strides our work represents in the field.

Fig. R1. EELS results of the CeCoIn₅/CeRhIn₅ superlattice showing well-defined interfaces (Naritsuka et al., PRL **120**, 187002 (2018)).

Fig. R2. STM topographic image of the atomically flat surface of CeCoIn₅ without texturing (L. Peng *et al.*, PRB **107**, L041101 (2023)). The height difference between top-left and bottom-right layers corresponds to 1 UCT of CeCoIn₅. The scale bar denotes 40 nm.

4. Inclusion of RBS data on stoichiometry would be insightful, particularly considering the lower T_c of the Tri-color film compared to stoichiometric CeCoIn₅. Exploring why the T_c is reduced in tri-layer films could provide valuable insights.

We agree that inclusion of RBS would be beneficial to check the stoichiometry. However, it is quite time consuming as the equipment to make this measurement is not available in the universities and laboratories in our area. We did, however, perform EDX and EELS. In addition, we examined the surface of CeCoIn₅ thin films using STM; The STM analysis revealed that the surface is exceptionally flat at the atomic level, with negligible impurity presence. From these observations, we conclude that the films are stoichiometric.

The observed decrease in T_c for superconductivity can be attributed to various factors, among which the band structure plays a pivotal role. In its bulk form, this system exhibits an anisotropic 3D Fermi surface, accompanied by a 3D antiferromagnetic (AFM) Q-vector, both of which are crucial for the emergence of unconventional superconductivity. However, the two dimensionalization leads to a notable alteration in the band structure and the AFM interactions. The change from 3D to 2D significantly influences the T_c , underscoring the critical interplay between dimensionality and superconductivity. Although a detailed exploration of these band structure changes is beyond the scope of our current study, we acknowledge its importance and have highlighted this aspect in the manuscript.

5. The conclusion that the asymmetric vortex motion perpendicular to the 2D plane is independent of in-plane current and magnetic fields needs further explanation. Why does vortex motion not vary in different crystallographic directions?

The intrinsic relationship between vortex motion in superconductors and Fermi velocity stems from the fact that the electronic structure of a vortex is dictated by its coherence length, which is, in turn, determined by the Fermi velocity. Additionally, the upper critical field H_{c2} is strongly influenced by this coherence length. Thus, by analyzing the anisotropy of H_{c2} within the plane, we can deduce the corresponding anisotropy—or lack thereof—of the Fermi velocity in the same plane.

Our experimental data demonstrate that H_{c2} is largely independent of the in-plane angle (Fig. R3). This observation leads us to infer that the average Fermi velocity within the plane is nearly isotropic. Consequently, we propose that the magnetic field exerts an isotropic influence within this plane. The isotropy is further mirrored in the behavior of the electrical current, which is also contingent on the Fermi velocity. The uniformity of average Fermi velocity within the plane indicates that the current's contribution is isotropic as well.

These findings lead us to conclude that the direction of vortex motion is independent of both the current and the magnetic field within the plane. This claim gains strength from data collected at high temperatures and low magnetic fields, conditions under which vortex dynamics are especially prominent. The H_{c2} versus T data for both nodal and antinodal configurations show remarkable similarity, reinforcing the isotropic nature of vortex contributions under these conditions.

Fig. R3. H_{c2} vs T plot for two magnetic-field orientations.

6. In Fig. 3, the reason behind the changes in peak position (R_{2w}/R_n) with different temperatures should be discussed. The reasons behind multiple oscillations in the antinode and less prominent oscillation in the node, especially at lower temperatures, should be addressed.

Regarding the position of the peak, it varies as expected due to the decrease in H_{c2} with increasing temperature. This is a consistent trend observed in our data. Concerning the dip observed around the peak of the node, we affirm that this is a genuine signal and not an artifact, as evidenced by its reproducibility in multiple experimental runs. This lends credibility to its significance in our findings. Regarding the oscillation highlighted by the reviewer, initially, we considered it an experimental noise owing to its subtle nature. However, upon further consideration prompted by the reviewer's insights, we acknowledge the possibility of an underlying structure. Recognizing the potential importance of this observation, we plan to investigate it thoroughly in future studies. We have added a sentence to touch this point in the revised manuscript.

7. Figure S5 in the supplementary information compares the non-reciprocal effect between tricolor and bicolor samples for specific configurations (when the magnetic field and the current are applied parallel to the nodal direction ($H \parallel [110]$, $I \parallel [1-10]$)). It would be beneficial to extend this comparison to the antinodal configuration ($H \parallel [100]$, $I \parallel [010]$) for a more comprehensive understanding.

At the suggestion by the reviewer, we performed additional measurements. Figure 4 in the revised manuscript depicts the Non-Reciprocal Electron Transport (NRET) of bicolor sample for nodal ($H \parallel [110]$, $I \parallel [1-10]$) and antinodal ($H \parallel [100]$, $I \parallel [010]$) configurations, indicated by red upper triangles and black open circles, respectively. For comparison, NRET of tricolor sample for antinodal configuration is also shown by upper blue circles. In the bicolor sample, the NRET is absent or negligibly small compared to that of the tricolor sample and no discernible difference is observed between the antinodal and nodal configurations. We thank the referee for highlighting this point.

Reviewer #2 (Remarks to the Author):

This manuscript describes the measurements on the second harmonic generation of the superconducting lattices, with a nice theory for the observed results. However, one puzzle prevents me from recommending the publication of the current form of this manuscript. According to my understanding, measurements on the second harmonic generation should be done when the resistance is finite, namely in the fluctuating regime of superconductivity. It occurs to me that the upper critical field here is determined by $0.5R_n$ criterion where normal state does not completely vanish. I am wondering, what would be H_{c2} when the criterion is 0 or $0.1R_n$? I would like to know how much the contribution of normal state is, since the normal state already breaks inversion symmetry and is sufficiently able to generate second harmonics.

We thank the referee for highlighting this important point of our study. The referee points out that the contribution to nonreciprocal electron transport effect (NRET) from a sample in the normal state should be considered, given that spatial inversion symmetry is broken even in this state. We can confidently conclude that the NRET observed in our study does not originate from the normal state. This is evident from the fact that in the normal state well above T_c , or well above H_{c2} where superconductivity is completely suppressed by the magnetic field, the second harmonic contribution becomes negligible, as clearly depicted in Fig. 3 in the main text and Fig. R4. This indicates that the non-reciprocal transport phenomena caused purely by the supercurrents that directly couple to the finite momentum Cooper pairs. It is important to clarify why the nonreciprocal signal is strongly suppressed in the normal state, despite the broken inversion symmetry. This suppression can be attributed to the fact that the signal is proportional to $1/E_F$ in the normal state and to $1/\Delta$ in the superconducting state, where E_F and Δ represent the Fermi energy and superconducting gap, respectively. As a result, the signal is significantly smaller in the normal state. We have explicitly state this in the revised manuscript.

In the context of magnetic field-induced resistive transitions in superconductors, particularly when the transition is broadened due to superconducting fluctuations as observed in this study, there is a degree of uncertainty in selecting the appropriate resistance threshold (0.9 R_n , 0.5 R_n , or 0.1 R_n) to define the mean-field upper critical field H_{c2} for superconductivity. However, given that NRET is vanishingly small in the normal state, it is natural to associate the mean field upper critical field with the emergence point of observable NRET. In the present case, the magnetic field corresponding to 0.5 R_n closely aligns with this emergence point. Consequently, this value has been adopted to define the mean-field upper critical field. Following a recommendation by a reviewer, H_{c2} was alternatively defined at a resistance of 0.1 R_n as shown in Fig. R4. This highlights that the intrinsic contribution of NRET predominantly becomes discernible in the resistance range between $R=0.1 R_n$ and $R=0.5 R_n$. We have explicitly state this point in the revised manuscript.

Fig. R4. The dashed line indicates the H_{c2} defined by $R(H_{c2}) = 0.1 R_n$.

Reviewer #1 (Remarks to the Author):

The authors have satisfactorily responded to my questions and implemented constructive revisions to the manuscript. Although not all my initial concerns were addressed, the final product demonstrates significant improvement. Based on this progress, I recommend publication.

Reviewer #2 (Remarks to the Author):

I think the reply and the revised manuscript are satisfactory for me, and my concerns are well addressed. I recommend the publication of this manuscript on Nature Communications.